# Revised Exon Structure of l-DOPA Decarboxylase (*DDC*) Reveals Novel Splice Variants Associated with Colorectal Cancer Progression

**DOI:** 10.3390/ijms21228568

**Published:** 2020-11-13

**Authors:** Pinelopi I. Artemaki, Maria Papatsirou, Michaela A. Boti, Panagiotis G. Adamopoulos, Spyridon Christodoulou, Dido Vassilacopoulou, Andreas Scorilas, Christos K. Kontos

**Affiliations:** 1Department of Biochemistry and Molecular Biology, Faculty of Biology, National and Kapodistrian University of Athens, 15701 Athens, Greece; partemaki@biol.uoa.gr (P.I.A.); maro.papatsirou@gmail.com (M.P.); michaelaboti2008@hotmail.com (M.A.B.); padamopoulos@biol.uoa.gr (P.G.A.); didovass@biol.uoa.gr (D.V.); ascorilas@biol.uoa.gr (A.S.); 2Fourth Surgery Department, National and Kapodistrian University of Athens, University General Hospital “Attikon”, 12462 Athens, Greece; spyridon.christodoulou@yahoo.gr

**Keywords:** l-aromatic amino acid decarboxylase (AADC), next-generation sequencing (NGS), transcriptomics, alternative splicing, protein isoforms, colon carcinoma, molecular biomarkers, biogenic amines, dopamine, serotonin

## Abstract

Colorectal cancer (CRC) is a highly heterogenous malignancy with an increased mortality rate. Aberrant splicing is a typical characteristic of CRC, and several studies support the prognostic value of particular transcripts in this malignancy. l-DOPA decarboxylase (DDC) and its derivative neurotransmitters play a multifaceted role in physiological and pathological states. Our recent data support the existence of 6 *DDC* novel exons. In this study, we investigated the existence of additional *DDC* novel exons and transcripts, and their potential value as biomarkers in CRC. Next-generation sequencing (NGS) in 55 human cell lines coupled with Sanger sequencing uncovered 3 additional *DDC* novel exons and 20 splice variants, 7 of which likely encode new protein isoforms. Eight of these transcripts were detected in CRC. An in-house qPCR assay was developed and performed in TNM II and III CRC samples for the quantification of transcripts bearing novel exons. Extensive biostatistical analysis uncovered the prognostic value of specific *DDC* novel exons for patients’ disease-free and overall survival. The revised *DDC* exon structure, the putative protein isoforms with distinct functions, and the prognostic value of novel exons highlight the pivotal role of DDC in CRC progression, indicating its potential utility as a molecular biomarker in CRC.

## 1. Introduction

Globally, colorectal cancer (CRC) is the third most commonly diagnosed malignancy and the second leading cause of death from cancer. An early stage diagnosis ameliorates the successful excision of the tumor; however, the appearance of CRC symptoms at an advanced stage and the low number of screening tests impede the timely diagnosis. Advancements in understanding the pathophysiological background have increased the effective treatment options for local and advanced disease, leading to the establishment of personalized treatment plans [1]. Especially for patients in tumor, node, and metastasis (TNM) stage III, the adjuvants that are currently administered are quite effective (most commonly 5-fluorouracil combined with oxaliplatin). On the other hand, the decision regarding the administration of chemical regimens on patients in TNM stage II is controversial. The presence of traditional “high-risk” pathologic factors in stage II colon cancer, including T4 stage (represents tumor size) and venous and lymphatic invasion, can identify a subgroup of patients with a recurrence risk approximating stage III disease, but not quite successfully [2]. Therefore, there is an urgent need to identify new molecular biomarkers that further define patient subsets in stage II disease with a more aggressive tumor development.

The increased heterogeneity and multistep initiation of CRC encumber the aforementioned endeavor. Nevertheless, the adoption of novel technological approaches has assisted in the elucidation of the molecular background of the pathobiology of this malignancy and in highlighting the critical involvement of alternative splicing [3]. Alternative splicing is the molecular mechanism to which the structural variation of transcripts and proteome diversity can be mainly attributed. One of the major processes that assesses the products of alternative splicing, leading the defective ones to degradation, is the nonsense-mediated mRNA decay (NMD). More specifically, it recognizes splice variants with a premature termination codon (PTC) and eliminates them, since their potential translation could lead to the generation of truncated and possibly harmful protein isoforms [4]. Aberrant splicing is commonly implicated in cancer initiation and development, and takes place on a genome-wide scale; therefore, it is considered as one of the hallmarks of cancer [3,5].

Due to the critical role of alternative splicing in both physiological and pathological states, its mechanism of action and the derived alternative splice variants have been thoroughly investigated so far. In particular, the introduction of next-generation sequencing (NGS) in molecular biology has assisted in the discovery of hundreds of novel transcripts, starting a new era in transcriptomics [6]. This in-depth analysis can uncover low abundant splice variants, some of which have been characterized as cancer-specific [7].

l-DOPA decarboxylase (*DDC*) is located on chromosome 7p12.2–p12.1 [8], and traditionally encompasses 17 exons, including two alternative first exons and an alternative terminating exon 10, indicating the existence of two distinct 5′ and 3′ untranslated regions (UTRs), respectively [9,10,11,12]. The two alternative first exons are named L1 and N1 and were initially considered mutually exclusive, generating two tissue-specific alternative 5′ UTRs, given that the translation start codon resides on the second exon. More specifically, transcripts bearing exon N1 were found to be expressed only in neuronal tissues, while transcripts bearing exon L1 were detected in non-neuronal tissues [13]. However, this notion has been repeatedly confuted, since transcripts encompassing exon N1 have also been detected in non-neuronal tissues, including the lung, liver, kidney, placenta, and in leukocytes [14,15,16,17].

Since the initial discovery of *DDC*, multiple alternative splice variants of this gene have been identified, and several of them are predicted to possess an open reading frame (ORF) [11,18,19]. Interestingly, our research group recently discovered 6 novel exons of *DDC* that are less frequent than the already known ones, via an NGS analysis in 53 cancer cell lines and 10 novel splice variants also bearing these novel exons [20]. These findings change the known exon structure of *DDC* and pose several questions regarding the impact of the incorporation of these novel exons in mature transcripts on DDC function and on normal cellular function in general.

DDC is an enzyme that has attracted researchers’ interest due to the multifaceted implication of its own and the biogenic amines that it biosynthesizes in human pathological states. More specifically, DDC (EC 4.1.1.28) is a pyridoxal-5-phosphate (PLP)-dependent, homodimeric enzyme [21], which catalyzes the decarboxylation of 3,4-dihydroxy-l-phenylalanine (l-DOPA) and 5-hydroxy-l-tryptophan (5-HTP) to dopamine and serotonin, respectively. There is evidence that DDC is also involved in the biosynthesis of other biogenic amines through the decarboxylation of other amino acids, including tryptophan, histidine, and phenylalanine. Due to this fact, DDC has also been termed aromatic l-amino acid decarboxylase (AADC) [22,23]. Additionally, it is involved in the biosynthetic pathways of monoamine neurotransmitters, including the biosynthesis of trace amines [24,25]. Therefore, ambiguity regarding the specificity of the enzyme’s substrate exists.

Besides its implication in biosynthetic pathways, recent data support the interaction of DDC with phosphatidylinositol 3-kinase (PI3K), and propose apoptotic and anti-proliferative functions of DDC [26]. The activation of the PI3K pathway and the inhibition of apoptosis are two properties that accompany malignant cells and lead to cancer (and especially CRC) initiation and progression [27,28]. Additionally, DDC is considered as a general biomarker for neuroendocrine malignancies [29], while *DDC* mRNA expression levels have proved to have prognostic value in several malignancies [30,31,32,33], including CRC [34], prostate cancer [35,36], stomach cancer [37], and gastric cancer [38]. These findings support that the implication of DDC in tumorigenesis is interesting and multifaceted, and encouraged us to investigate the alternative splicing pattern of *DDC* in CRC.

Prompted by the discovery of the aforementioned 6 *DDC* novel exons, we decided to further investigate their expression pattern in CRC cell lines and query the existence of other *DDC* novel exons in malignant conditions. For this purpose, we conducted an NGS analysis coupled with Sanger sequencing and discovered 3 additional novel exons and 20 novel splice variants of *DDC*. Seven of these 20 transcripts are expected to possess an ORF and to encode peptides. Out of the 9 novel exons, 4 were found to be highly expressed in CRC cell lines. For this reason, they were selected for quantification of their expression levels via an in-house-developed qPCR assay, involving 109 CRC samples from patients in TNM stages II and III and 40 adjacent normal ones. The present study focused on CRC TNM stage II and stage III patients due to the inefficiency of the current TNM staging to predict the survival outcome of CRC patients classified in these subgroups. Following an extensive biostatistical analysis, it was shown that the increased expression levels of three novel exons had an impact on patients’ overall survival (OS) and disease-free survival (DFS). In addition, these novel exons were able to further stratify patients of distinct TNM stages in subgroups with discrete disease progression.

## 2. Results

### 2.1. Discovery of Novel Exons and Novel Splice Sites of the DDC Gene

The NGS data analysis with the use of our in-house-developed algorithm [39] revealed the existence of 3 *DDC* novel exons, which are incorporated in novel splice variants of *DDC* that are less frequent than the already annotated ones (Appendix A). These 3 *DDC* novel exons and the 6 *DDC* novel exons that were discovered in our previous research study [20] were termed X1 to X9; X2, X7, and X9 were the 3 exons derived from the present study. Exon X7 was considerably less abundant compared to the rest. Exon X9 is a 5′ extension of exon 10alt, which exists in the *alt-DDC* transcript (accession number: NM_001242890.2, encodes DDC is.6) and shares 13 nucleotides at its 3′ end with the annotated exon 10alt. Additionally, 2 novel splice sites of the previously discovered exon X3 were identified in the present study, generating 2 shorter versions of exon X3. More specifically, the first one has a length of 60 nucleotides (nt) and is termed exon X3a, while the latter has a length of 9 nt and is named exon X3b. These 2 exons were not further investigated in the present study.

### 2.2. Identification of DDC Novel Splice Variants

From the expression analysis in the 11 cDNA pools of distinct cancer origin, 3 cDNA pools, originating from CRC, hepatocellular, and lung cancer cell lines, were the ones in which most of these 9 novel exons were observed under the present experimental conditions (Figure 1). Interestingly, novel exon X7 was not detected in any of these 11 cDNA pools, probably due its low frequency rate.

Sanger sequencing of the most intense gel bands and assembly of the respective sequences led to the determination of the exon structure of 20 *DDC* novel splice variants. The exon structure of these novel transcripts and their respective accession numbers are depicted in Figure 2. These transcripts all include one of the novel exons and are characterized by the full known exon structure, with or without exon 3, which is considered one of the most frequent splice variations in *DDC* pre-mRNA. Four of these splice variants (GenBank^®^ accession numbers: MN447448.1, KP729825.2, KP729826.2, and MW092772.1), which contain novel exons X1, X3, and X6, respectively, are characterized by the second known 5′ UTR (i.e., exon N1 as the first exon).

Particularly in the CRC cDNA pool, 4 out of the 9 novel exons were observed to be more abundant, namely X1, X3, X8, and X9. The other 5 novel exons were either detected in low abundance or not detected at all with this particular experimental protocol in the CRC cDNA pool. Based on this finding, only the aforementioned 4 novel exons were further investigated in the 7 CRC cell lines. However, the possibility that the other 5 novel exons are expressed in specific CRC cell lines should not be excluded. The results derived from the CRC cDNA pool, in conjunction with the results from the expression analysis of the 7 CRC cell lines, showed the existence of 8 out of 20 novel splice variants in CRC cell lines, namely MN447447.1, MN447437.1, MN447450.1, MN447438.1, MN447444.1, MN447456.1, MN447446.1, and MW092775.1.

Interestingly, following the expression analysis of the 7 CRC cell lines, an expression pattern of the novel exons arose as well. More specifically, all these novel transcripts were observed in the Caco-2 cell line. Transcripts encompassing exons X1, X3, and X8 were highly expressed in the COLO-205 cell line as well. These findings are in accordance with the results from the real-time qPCR assays, in which Caco-2 and COLO-205 were the two cell lines with higher expression of the studied novel exons. Exon X8 was highly expressed in the HT-29 cell line as well.

This abundant expression of exons X1, X3, X8, and X9 advocated for their selection regarding their further investigation in tissue samples.

### 2.3. Deduced DDC Protein Isoform Sequences and Predicted 3D Structure Models 

Seven out of 20 novel transcripts have an ORF and are predicted to encode new protein isoforms. V.32 and v.33 possess the same ORF and likely encode the protein isoform is.14, as they differ in their 5′ UTR. Is.14, is.15, is.16, is.17, is.18, and is.19 are considered as potential novel protein isoforms of DDC, while the lengths of their amino acid sequences are 521 aa, 483 aa, 511 aa, 473 aa, 513 aa, and 475 aa, respectively. Most of the deduced protein isoforms are larger compared to the main DDC isoform (is.1) at 480 aa residues, to DDC is.2 at 442 aa residues (the internal peptide encoded by exon 3 is absent), and to DDC is.6 at 338 aa residues. Moreover, they are characterized by different molecular weight and isoelectric point (pI) values in comparison to the main isoform (Table 1). In CRC cell lines, 2 of the 8 observed transcripts were shown to possess an ORF. Therefore, 2 novel protein isoforms, namely is.14 and is.15, are predicted to be expressed in CRC.

The 3D protein structures with the highest confidence score were built by the I-TASSER server (https://zhanglab.ccmb.med.umich.edu/I-TASSER/) and they demonstrate that all 6 novel protein isoforms contain a distinct combination of α-helixes, β-sheets, and coils compared to DDC is.1, is.2, and is.6 (Appendix A). Interestingly, it was predicted that these deduced protein isoforms can have additional ligands besides PLP. The ligands with the highest confidence score were carbidopa, a well-known inhibitor of DDC function [40], leucine, and glycine, which bind to specific residues of distinct protein isoforms (Figure 3). The ligands carbidopa and glycine also bind to DDC is.6 and is.1, respectively.

These 7 deduced protein isoforms contain one additional peptide deriving from each novel exon compared to the traditional protein isoform. These deduced peptides can alter the properties of the respective protein and also possess additional sites for post-translational modifications (Table 1). The most common additional modifications are phosphorylation and S-palmitoylation.

### 2.4. Clinical and Biological Characteristics of the CRC Patients

For this research study, 109 cancerous and 40 paired non-cancerous colorectal tissue samples were collected from 109 patients with primary CRC. Among the 109 patients, 62 were men and 47 were women. The median age of the CRC patients was 68 years (35–93). With regard to the tumor histological grade, 8 patients were diagnosed with grade I (well-differentiated), 84 with grade II (moderately differentiated), and 17 with grade III malignant tumors (poorly differentiated), based on the WHO classification system. Moreover, 63 malignant lesions were classified in TNM stage II and 46 in TNM stage III; a more thorough description of the patients’ clinical and biological features is presented in Table 2.

### 2.5. Reduced Expression Levels of the DDC Exon X1 in CRC Samples Compared to Adjacent Normal Tissue, and Downregulation of the DDC Exons X1, X8, and X9 During the Transition from TNM II to TNM III 

Based on the abundance of *DDC* exons X1, X3, X8, and X9 in the CRC cDNA pool, the expression of these 4 novel exons was next quantified in human tissue samples. The median *DDC* exon X1, X3, X8, and X9 expression levels were equal to 0.31, 0.32, 0.20, and 0.017 relative quantification units (RQU) in tumor samples, respectively; while in non-cancerous samples the median *DDC* exon X1, X3, X8, and X9 expression levels were 1.05, 0.47, 0.001, and 0.001 RQU, respectively (Table 2). This decrease in *DDC* exon X1 expression in CRC tissue specimens compared to the 40 paired non-cancerous ones is profound, as clearly illustrated in Figure 4 (*p* = 0.018). In fact, in 27 out of the 40 sample pairs, the *DDC* exon X1 levels were higher in the normal tissue samples in comparison with the cancerous one, while the opposite observation was made for the other 13 pairs of tissue samples. Following the classification of the tumors according to their TNM stage, a decrease in the mean *DDC* exon X1 expression was observed in parallel with the progression from TNM II to TNM III (*p* = 0.005) (Appendix A). Therefore, *DDC* exon X1 tends to be suppressed in malignant tumors.

The expression levels of the other three *DDC* novel exons did not appear to have statistically significant differences between cancer and adjacent normal pairs. However, *DDC* exon X8 and *DDC* exon X9 expression is associated with TNM stage progression. More specifically, the comparison of *DDC* exon X8 and X9 expression levels between TNM stages II and III indicates a negative association with TNM stage progression (*p* = 0.003, *p* = 0.009, respectively) (Appendix A).

The conceivable difference between TNM stages II and III is the nodal metastasis status (N), which is characterized as N0 (no nodal metastasis) in TNM stage II and as N1 or N2 in TNM stage III. Therefore, the statistically significant mean *DDC* novel exon expression levels between these two stages is reflected in the different N statuses. Even though the aim of the study is not to define a cause and effect relation, the lack of association between the expression of these novel exons and the T status supports that the association of *DDC* novel exons expression levels and TNM stage could be related to the N status.

### 2.6. DDC Novel Exons Expression Status and Association with Clinicopathological Characteristics of CRC 

In order to classify the expression statuses of all 4 *DDC* novel exons of each tissue sample as either positive or negative, we determined the optimal cut-off value (see Materials and Methods). Concerning *DDC* exon X1 expression levels, 49.5% of the CRC patients were characterized as *DDC* exon X1-negative, with relative *DDC* exon X1 expression being lower than 0.31 RQU, whereas the other 50.5% of CRC patients were marked as *DDC* exon X1-positive. Moreover, 49.5% of the CRC patients were characterized as *DDC* exon X3-negative, with relative *DDC* exon X3 expression being lower than 0.32 RQU, whereas the other 50.5% of CRC patients were marked as *DDC* exon X3-positive. Regarding *DDC* exon X8 expression levels, 83.5% of the CRC patients were characterized as *DDC* exon X8-negative, with relative *DDC* exon X8 expression being lower than 4.5 RQU, whereas the other 16.5% of CRC patients were marked as *DDC* exon X8-positive. Lastly, 63.3% of the CRC patients were characterized as *DDC* exon X9-negative, with relative *DDC* exon X9 expression being lower than 0.24 RQU, whereas the other 36.7% of CRC patients were designated as *DDC* exon X9-positive.

The association of patients’ *DDC* novel exons expression status with distinct clinicopathological features, including TNM stage and histological tumor stage, was investigated. However, no significant associations were detected between the expression status of each *DDC* novel exon and the clinical parameters studied.

### 2.7. DDC Novel Exon Differential Expression is an Independent Predictor of CRC Patients’ Relapse

The univariate Cox regression analysis revealed the prognostic significance of *DDC* exons X3, X8, and X9 expression and of the tumor site for patients’ relapse (Table 3). The significance of the tumor site was maintained in the multivariate Cox regression analysis as well. More precisely, it was observed that CRC patients with a *DDC* exon X3-positive expression status had a lower risk of tumor recurrence than the *DDC* exon X3-negative ones (HR = 0.45, 95% CI = 0.20–0.91, *p* = 0.033) (Table 3). The significantly higher DFS of *DDC* exon X3-positive CRC patients as compared to those who were *DDC* exon X3-negative is clearly depicted in the Kaplan–Meier curves (*p* = 0.043) (Figure 5A). The prognostic value of *DDC* exon X3 expression status remained unaffected in the multivariate Cox regression analysis, and therefore it could be considered as an independent biomarker (HR = 0.42, 95% CI = 0.14–0.82, *p* = 0.037) (Table 3).

Concerning *DDC* exon X8 expression status, an association opposite from the aforementioned one emerged. More specifically, CRC patients bearing *DDC* exon X8-positive tumors were characterized by a significantly shorter DFS time interval compared to those with tumors expressing minimal or no *DDC* exon X8 (HR = 2.51, 95% CI = 1.00–5.69, *p* = 0.010) (Table 3). This differential patient relapse time depending on the *DDC* exon X8 expression status is also depicted in the Kaplan–Meier curves (*p* = 0.015) (Figure 5B). The prognostic value of *DDC* exon X8 expression status was retained in the multivariate Cox regression analysis, designating this parameter as a potential unfavorable independent prognostic indicator of tumor recurrence (HR = 3.17, 95% CI = 1.05–14.93, *p* = 0.011) (Table 3). However, due to the widened 95% CI, we should be moderate regarding our conclusions.

Finally, elevated *DDC* exon X9 expression levels were proven to act as a favorable indicator of CRC patients’ DFS (HR = 0.32, 95% CI = 0.097–0.64, *p* = 0.015) (Table 3). The higher DFS time interval of CRC *DDC* exon X9-positive patients compared to the negative ones is also depicted in Kaplan–Meier curves (*p* = 0.014) (Figure 5C). The significance of the *DDC* exon X9 expression status in the prognosis of the patient’s DFS time period was sustained in the multivariate Cox regression analysis, as well (HR = 0.32, 95% CI = 0.10–0.60, *p* = 0.009) (Table 3). Thus, the *DDC* exon X9 expression status in CRC samples constitutes an independent prognostic biomarker for patients’ relapse.

### 2.8. DDC Novel Exon Differential Expression is an Independent Predictor of CRC Patients’ OS

Consistently, univariate Cox regression analysis revealed the prognostic significance of the expression status of *DDC* exons X3, X8, and X9 and of the tumor site for CRC patients’ OS (Table 4). However, the prognostic value of the tumor site is marginally insignificant in multivariate Cox regression analysis. In the present study, CRC patients bearing *DDC* exon X3-positive tumors succumbed after a longer OS time interval in contrast with those with tumors expressing minimal or no *DDC* exon X3 (HR = 0.42, 95% CI = 0.15–0.87, *p* = 0.031). The favorable impact of *DDC* exon X3-positive expression status on CRC patients’ OS is also illustrated in the Kaplan–Meier curves (*p* = 0.033) (Figure 5D). In the multivariate Cox regression analysis, the significance of the *DDC* exon X3 expression status in the prognosis of the patients’ OS time period remained unaffected (HR = 0.31, 95% CI = 0.095–0.61, *p* = 0.016) (Table 4). Therefore, the *DDC* exon X3-positive expression status in CRC could be characterized as a favorable independent prognostic value for OS.

On the contrary, CRC patients with high *DDC* exon X8 expression status are accompanied by decreased OS time interval compared to the ones who are characterized by low *DDC* exon X8 expression levels (HR = 2.60, 95% CI = 1.16–5.84, *p* = 0.006) (Table 4). This tendency is also observed in the Kaplan–Meier curves (*p* = 0.010) (Figure 5E). High *DDC* exon X8 expression levels could be considered as an independent unfavorable prognostic indicator for OS of CRC patients, since statistically significant values were preserved in the multivariate Cox regression analysis as well (HR = 3.63, 95% CI = 1.18–32.25, *p* = 0.011) (Table 4). However, we should be moderate regarding our conclusions on account of the widened 95% CI.

Regarding the association of *DDC* exon X9 expression status with patients’ OS, the same tendency was observed as with patients’ DFS. Particularly, *DDC* exon X9-positive patients are characterized by a higher OS time period than the negative ones (HR = 0.26, 95% CI = 0.061–0.55, *p* = 0.012) (Table 4). This prognostic outcome is also obvious in Kaplan–Meier curves *(p* = 0.007) (Figure 5F). The fact that the prognostic significance of *DDC* X9-positive status was maintained in the multivariate Cox regression analysis highlights its potential independent value as a molecular biomarker (HR = 0.23, 95% CI = 0.078–0.42, *p* = 0.004) (Table 4).

### 2.9. Expression Levels of the DDC Novel Exons in CRC Tissues after Patient Stratification according to Established Prognosticators are Prognostic Factors for DFS and OS

The high importance of the tumor site, histological grade, and TNM staging in the prediction of DFS and OS intervals in patients with CRC led to the stratification of CRC patients according to these factors.

Firstly, CRC patients were stratified based on their TNM stage in two groups, so that the impact of the *DDC* novel exons expression status on DFS and OS of CRC patients was assessed. It was observed that *DDC* exon X8-positive patients with TNM stage II malignant lesions have significantly lower DFS and OS time intervals than the *DDC* exon X8-negative ones (*p* = 0.034 for DFS Kaplan–Meier curve; *p* = 0.008 for OS Kaplan–Meier curve) (Figure 6A,B). Furthermore, *DDC* exon X9-positive patients with TNM stage II malignant lesions were characterized by lower probability to relapse or succumb compared to the *DDC* exon X9-negative ones (*p* = 0.049 for DFS Kaplan–Meier curve; *p* = 0.019 for OS Kaplan–Meier curve) (Figure 6C,D). Finally, positive *DDC* exon X3 expression levels in the tumor could serve as a favorable prognostic indicator in patients with TNM stage III tumors, as the patients with *DDC* exon X3-positive tumors had significantly higher DFS and OS rates in comparison with those bearing *DDC* exon X3-negative tumors (*p* = 0.015 for DFS Kaplan–Meier curve; *p* = 0.011 for OS Kaplan–Meier curve) (Figure 6E,F).

Secondly, a stratification followed of the CRC patients into two distinct groups according to their tumor site. Even though *DDC* exon X8 expression status was not associated with patients’ DFS in the distinct subgroups (Appendix A), *DDC* exon X8-positive patients with colon tumors were more likely to pass away compared to the *DDC* exon X8-negative ones (*p* = 0.010) (Appendix A). On the other hand, *DDC* exon X9-positive patients with colon tumors were less likely to relapse and pass away, in contrast with the *DDC* exon X9-negative ones (*p* = 0.023 for DFS Kaplan–Meier curve; *p* = 0.006 for OS Kaplan–Meier curve) (Appendix A).

Thirdly, possible impacts of the *DDC* novel exon expression status on DFS and OS were sought, following the stratification of CRC patients based on their tumor histological grade. Among the patients with grade II tumor lesions, those with *DDC* exon X8-positive tumors had a higher probability of inferior DFS and OS in comparison with those bearing *DDC* exon X8-negative tumors (*p* = 0.047 for DFS Kaplan–Meier curve; *p* = 0.030 for OS Kaplan–Meier curve) (Appendix A). Additionally, *DDC* exon X9-positive patients with grade II tumor lesions were characterized by a non-statistically significant probability of relapse but with a lower probability to succumb compared to the *DDC* exon X9-negative ones (*p* = 0.064 for DFS Kaplan–Meier curve, *p* = 0.033 for OS Kaplan–Meier curve) (Appendix A). In patients with grades I and III malignant lesions, the association of *DDC* novel exon expression level with DFS and OS could not be evaluated because of the limited number of patients.

## 3. Discussion

The dysregulation of splicing is a general feature of cancer. Particularly in CRC, several genes are alternatively spliced and give rise to specific transcripts that have been associated with the etiology of this malignancy. Exceptional examples are *APC*, *KRAS*, and *TP53*, whose splice variants have distinct and even antagonistic functional properties [3]. However, most research studies focus on the alternative splicing of known cancer-critical genes, neglecting that this phenomenon takes place on a genome-wide scale.

Despite the great progress that has been achieved towards the elucidation of the role of alternative spliced products in cancer, several questions remain unanswered. The emergence of high-throughput NGS analysis enables the detailed investigation of transcriptomes in tumors, and can assist towards the understanding of the contribution of alternative splicing in cancer initiation and progression [41]. However, high-throughput sequencing poses several challenges as well. One of these is that the analysis of differential splicing among samples or cell lines is biased towards the more abundant transcripts, and it is, thus, highly sensitive to the sequencing depth [42]. In the present study, an in-house-developed algorithm for the analysis of the NGS data was implemented, which assisted in the discovery of these lower abundant novel exons and novel splice sites due to its high sensitivity [39]. Overall, 3 *DDC* novel exons emerged from this NGS data analysis.

The expression analysis in the cDNA pools of distinct cancer origin revealed the exon structure of 20 novel transcripts, each comprising one novel exon, namely the aforementioned novel exons X2 and X9 and the 6 novel exons that emerged from our previous research study [20]. Most of these exons, and subsequently these transcripts, are mainly expressed in colorectal, hepatocellular, and lung cancer cell lines. Interestingly, two of the most common metastases from CRC are the ones in the lungs and liver; hence, the similar alternative splicing patterns could be partly attributed to this relation among the three malignancy types. However, further investigation regarding this notion is required. Furthermore, similar splicing patterns between CRC cell lines and a lung cancer cell line were also observed, as can be expected considering their similar tissue of origin [5].

Seven of the novel transcripts possess an ORF and can be potentially translated, generating 6 novel protein isoforms. The in silico post-translational analysis of these deduced protein isoforms uncovered additional sites for post-translational modifications. These modifications can alter the tertiary protein structure, the protein affinity and specificity to its substrates, the protein localization, and interactions with other molecules. Interestingly, other ligands besides PLP are predicted to bind specifically to these deduced protein isoforms, while the affinity of each one to each protein is different. For instance, the well-known DDC inhibitor [40] carbidopa is predicted to bind with high confidence only to is.14 and is.16, a finding that raises questions regarding the regulation of DDC activity. A previously unaccounted interaction of these DDC protein isoforms with the amino acid leucine also arose from this investigation. Interactions with other ligands besides PLP may lead to conformational changes of DDC isoforms, and hence to their binding to different substrates. Therefore, the hypothesis regarding the different substrates and functions of the deduced protein isoforms is supported. This finding is quite interesting, taking into account the existing ambiguity regarding the substrate specificity and localization of the DDC enzyme [43]. Considering the decisive role of biogenic amines in human physiological functions, including neurotransmission, growth, and apoptosis, and that impairment in their metabolic pathways is related to many different pathological phenotypes, including cancer [44], further investigation of the aforementioned hypothesis could shed light on cancer pathobiology, potentially providing new therapeutic targets.

Particularly in CRC, biogenic amines such as serotonin and dopamine affect both the initiation and progression of this malignancy. More specifically, dopamine exerts an inhibitory impact on tumor growth and angiogenesis, while it increases the efficacy of anti-cancer drugs in CRC [45,46]. In contrast, serotonin plays a dual role in CRC; there are indications for both oncogenic and onco-suppressive roles of serotonin [47]. Since these two catecholamines are synthesized by DDC, the function of this enzyme in CRC is considered crucial. Given that our data support the existence of 2 potential protein isoforms in CRC cell lines, further investigation regarding their function would be interesting and beneficial for the elucidation of the multifaceted role of DDC in normal and pathological states.

The expression analysis of these novel exons and transcripts in CRC cell lines showed that their expression is higher in Caco-2, a cell line in an early-stage mutational status. Following the expression analysis of novel exons in CRC samples and their adjacent counterparts, a statistically significant difference between the *DDC* X1 expression in malignant tissue samples and their normal counterparts emerged. In detail, the majority of non-cancerous samples had an increased *DDC* X1 expression compared to the cancerous ones. Additionally, *DDC* X1 expression levels were significantly higher in patients with tumors in TNM stage II than in patients with tumors in TNM stage III. These findings indicate a potential decrease in the *DDC* mRNA levels in the transition from normal to cancerous states and in the progression of TNM staging, which could lead to a decrease in the respective protein levels as well. All these findings support a potential tumor-protective role of the transcripts containing X1 and of the protein isoforms that they likely encode, and encourage their further investigation in the context of biosynthesis of biogenic amines that are beneficial to cancer treatment.

The other novel transcripts did not possess an ORF and could be considered candidates for NMD decay. However, the NMD mechanism is frequently deregulated in cancerous states. So far, several studies have reported that cancer tissues are more prone to splicing noise than healthy ones, which is supported by the fact that cancer-specific transcripts more frequently contain premature stop codons and are less abundant than normal transcripts in general [48]. Therefore, distinct splicing variants can serve as biomarkers, and this potential is supported by several studies. Among the most well-known examples is *CD44* [49]. Particularly in CRC patients, low expression of the *CD44V6* transcript has been associated with improved survival [50].

Prompted by this potential of splice variants, we investigated the value of the specific *DDC* novel exons as biomarkers, and specifically of those that were observed to be more abundant from the expression analysis in the cDNA pools. Their utility as biomarkers was evaluated in CRC tissue samples; however, due to the emerging value of liquid biopsy and its daily use in clinical practice, it would be interesting to evaluate these *DDC* novel exons in the serum or plasma of CRC patients. Following the extensive biostatistical analysis, it arose that *DDC* exons X3, X8, and X9 have a potential prognostic value. More specifically, *DDC* exon X3- and X9-positive CRC patients have lower probability to relapse and succumb than the negative ones, designating these characteristics as favorable prognostic biomarkers for both DFS and OS. On the other hand, high *DDC* X8 expression levels can serve as an adverse prognostic indicator for both DFS and OS. The patients involved in the present study are characterized by TNM stages II and III tumors, and as previously mentioned, the existing TNM staging fails to distinguish the progression between TNM II and III patients. Hence, the urgent need for biomarkers that efficiently predict the development of the disease in these two stages points out the importance of our findings.

The stratification of the patients into two distinct categories based on their TNM stage strengthened the aforementioned conclusion regarding the prognostic utility of these novel exons expression levels. More specifically, *DDC* X9-positive patients with TNM stage II tumors have increased DFS and OS time intervals compared to the negative ones, while *DDC* X8- positive patients with TNM stage II tumors display the opposite outcome. Interestingly, the *DDC* X9-negative and *DDC* X8-positive ones were characterized by a similar or even worse prognosis compared to patients with TNM III tumors. These findings are consistent with the statistically significant difference in novel exon expression regarding the N status and the decreased expression during TNM stage progression. Additionally, high *DDC* X3 expression levels in patients with TNM stage III malignant lesions were associated with higher DFS and OS time periods than the negative ones, supporting the existence of a subgroup inside TNM III patients with better survival rates who could follow a different treatment approach.

Since our knowledge regarding the molecular background of CRC is evolving and the significance of global genomic and epigenomic statuses in the initiation and progression of this malignancy is undoubtful, molecular classification of CRC patients has been adopted in clinical practice. Particularly, microsatellite instability (MSI) status, chromosomal instability (CIN) status, and CpG island methylator phenotype (CIMP) status play significant roles in determining clinical, pathological, and biological characteristics of CRC [51]. Therefore, it would be interesting to evaluate the potential of these *DDC* novel exons as biomarkers in subgroups of CRC patients with distinct genomic and epigenomic statuses.

The importance of these novel exons as potential prognostic biomarkers is supported by the outcomes of the Cox regression analysis, in which *DDC* exon X3, X8, and X9 emerged as independent indicators for both DFS and OS. Interestingly, *DDC* X9-positive CRC patients with tumors located in the colon were characterized by higher DFS and OS time intervals than the negative ones, while *DDC* X8-positive patients with the same tumor localization exhibited lower OS time periods than the negative ones. The existing literature suggests that cancer patients with colon-localized tumors have a shorter OS interval compared to the ones with rectum localized tumors [52]; therefore, the discovery of biomarkers able to predict the prognosis of patients inside this stratum is crucial, underlining the significance of our findings. Moreover, CRC patients with moderately differentiated tumors were further stratified in subgroups with distinct prognoses according to *DDC* X8 and X9 expression levels, accentuating the pluralistic value of these novel exons as CRC biomarkers.

The favorable or unfavorable prognostic value of these *DDC* novel exons can be interpreted from a molecular perspective as well. There are several studies that support the association of the NMD mechanism and alterative splicing. This combinational function alters the transcripts ratio and decreases the expression levels of mRNA molecules that encode peptides [53]. In a cancerous state, this altered ratio could favor or hinder tumor initiation and progression, depending on the impact of the proteins deriving from these mRNAs on cancer cells. Especially DDC is an enzyme that participates in the biosynthesis of biogenic amines that can facilitate or impede tumor progression, and hence a decrease in its mRNA levels can either have a favorable or adverse impact on cancer development. The increased cancer cell-to-cell variability in splicing compared to the minimal fluctuations in the ratios of transcripts in non-cancerous cells should be taken into consideration for the interpretation of the results [54]. The consequences of variability in the splicing ratio may be as significant as the ones of gene expression level fluctuation, which can affect several cellular behaviors [55].

The results from the present study pave the way for future investigation of the role of DDC in CRC and in malignancies in general. One interesting aspect that has not been examined yet is the potential crosstalk between the nervous and immune systems, which is orchestrated by the DDC function. There are studies that support the presence of enzymatically active DDC in human leukocytes, as well as in the histiocytic lymphoma cell line U-937 [15,16]. Additionally, dopamine, serotonin, and traces amines, which are the predominant substrates of DDC, can play a mediating role in immunological responses linked with pathological states, including cancer and diseases of the gastrointestinal track [56,57,58]. In the current study, the expression levels of *DDC* exons X1, X8, and X9 were differential among N status. Considering the critical roles of the immune system, lymph node metastasis, and neurotransmitter expression in cancer progression, a future investigation of DDC in this context would be intriguing.

## 4. Materials and Methods

An overview of the experimental workflow that was followed in the present study is presented in Figure 7.

### 4.1. Sample Collection and Storage

In total, 109 malignant colorectal tumors and 40 paired non-cancerous colonic tissue specimens were collected from patients with colorectal cancer, who were subjected to surgery at University General Hospital “Attikon”. Immediately after tumor resection, samples were snap-frozen in liquid nitrogen. Histological characterization of all tumors and adjacent non-cancerous colonic tissues was performed by a pathologist.

The clinicopathological features of CRC patients’ tumors included the primary tumor size and extent (T), regional lymph node count and metastasis (N), as well as the distant metastasis (M), histological grade, age, and gender. The inclusion criteria included TNM stages II or III, as well as complete follow-up information regarding DFS and OS. The median follow-up time was 52 months. This research study was conducted in accordance with the ethical standards of the 1964 Declaration of Helsinki and its later amendments. The research study was approved by the institutional Ethics Committee of the University General Hospital “Attikon” (approval number: 31; 29 January 2009). Moreover, written informed consent was obtained from all participants.

### 4.2. Culture of Human Cell Lines

In the present study, 53 human cancer cell lines originating from 17 distinct types of malignancies and 2 non-cancerous ones (1.2B4 and HEK-293) were cultivated (Appendix A). All cell lines were subcultured following the American Type Culture Collection (ATCC) guidelines.

### 4.3. RNA Extraction and cDNA Synthesis

Each sample was homogenized. Total RNA extraction from each of the 55 cell lines and the 149 patients’ samples followed using the TRIzol^®^ Reagent (Ambion™, Thermo Fisher Scientific Inc., Waltham, MA, USA) according to the manufacturers’ instructions. All RNA samples were diluted in DEPC-treated H_2_O and stored at −80 °C until further use. Each RNA sample was evaluated spectrophotometrically at 260 and 280 nm regarding its purity and concentration using the BioSpec-nano Micro-volume UV-Vis Spectrophotometer (Shimadju, Kyoto, Japan). A total of 5 μg of RNA from each cell line and 2 μg from each tissue sample were reverse transcribed. In addition to the RNA volume, each reaction mixture contained 1 μL of an oligo-dT-adaptor primer (50 μM) (5′-GCGAGCACAGAATTAATACGACTCACT ATAGGTTTTTTTTTTTTVN-3′, where V = G, A, C; and N = G, A, T, C), 1 μL 10 mM dNTP Mix (10 mM each dATP, dGTP, dCTP, and dTTP at neutral pH), and DEPC-treated H_2_O to a final volume of 10 μL. The mixture was heated to 65 °C for 5 min and then was quickly transferred onto ice for 1 min. Next, in each RNA sample, 1 μL of 5X First-Strand Buffer, 0.1 M DTT, 40 units RNaseOUT™ recombinant RNase inhibitor (Invitrogen™, Thermo Fisher Scientific Inc.), and 200 units of Reverse Transcriptase were supplemented, leading to a final reaction volume of 20 μL. For the reverse transcription of RNA deriving from cell lines, SuperScript^®^ III Reverse Transcriptase (Invitrogen™) was used, while for the RNA deriving from tissue samples, M-MLV Reverse Transcriptase (Invitrogen™) was used. Each final mixture originating from cell lines was incubated at 50 °C for 60 min, while each final mixture originating from tissues was incubated at 37 °C for 52 min. All reactions were inactivated at 70 °C for 15 min. The whole process was performed in an ABI 9700 PCR System (Applied Biosystems™, Thermo Fisher Scientific Inc.). The 55 first strand cDNAs, which were synthesized from the cancer cell lines, were diluted 1:5 in DEPC-treated H_2_O and then randomly mixed at equal volumes, generating 11 cDNA pools, each containing 5 cDNAs.

### 4.4. DDC Variant Amplification and Product Purification

The expressed *DDC* splice variants were specifically amplified via touchdown PCR and the use of 2 gene-specific primers. More precisely, the forward primer annealed to exon 2 prior to the known translation start codon, while the reverse primer specifically annealed to exon 14 at the end of the known coding sequence. All primers used in the present study were designed using Primer BLAST (Appendix A). Touchdown PCR was conducted for each of the 11 cDNA pools at a final mixture of 25 μL in a Veriti 96-Well Fast Thermal Cycler (Applied Biosystems™). Each reaction mixture included KAPA Taq Buffer A (Kapa Biosystems Inc., Woburn, MA, USA) with MgCl2 at a final concentration of 1.5 mM, 0.2 mM dNTPs, 0.4 μM of each primer, and 1 unit of KAPA Taq DNA Polymerase (Kapa Biosystems Inc.).The thermal protocol included the following steps: an initial denaturation step 95 °C for 3 min, 35 cycles of amplification consisting of: (1) a denaturation step at 95 °C for 30 s; (2) an annealing step for 30 s at an initial temperature of 65 °C that was progressively lowered by 0.2 °C/cycle, and an extension step at 72 °C for 2 min; and (3) a final extension step at 72 °C for 5 min.

Next, nested touchdown PCR was performed using an inner pair of primers, with the aim of increasing the sensitivity and specificity for the *DDC* gene. For this purpose, the 11 first-round PCR products were diluted and further used as templates for nested PCR. The thermal protocol and the reaction reagents were the same as in the first-round PCR.

Afterwards, purification of the 11 nested PCR products was performed with the use of NucleoSpin^®^ Gel and PCR Clean-up kit (Macherey-Nagel GmbH and Co. KG, Duren, Germany), according to the manufacturer’s protocol. The concentration and the quality of the purified PCR products were determined spectrophotometrically at 260 and 280 nm in a BioSpec-nano Micro-volume UV-Vis Spectrophotometer (Shimadju), prior to their storage at −20 °C until further use.

### 4.5. Semi-Conductor NGS and Data Analysis

Equal volumes of the purified PCR products were mixed. Next, 1 μg of pooled PCR products was used for NGS library preparation with the Ion Xpress™ Plus Fragment Library Kit (Ion Torrent™, Thermo Fisher Scientific Inc.) following the manufacturer’s standard protocol. The concentration of the DNA library was determined using the Ion Library TaqMan™ Quantitation Kit (Ion Torrent™) in an ABI 7500 Fast Real-Time PCR System (Applied Biosystems™). The DNA library was then used along with the Ion PGM™ Template OT2 400 Kit (Ion Torrent™) for NGS template preparation in an Ion OneTouch™ 2 System (Ion Torrent™). The NGS template quality was confirmed with the Ion Sphere™ Quality Control Kit in a Qubit^®^ 2.0 Fluorometer (Invitrogen™), before being enriched with the Ion PGM™ Template OT2 400 Kit (Ion Torrent™) in an Ion OneTouch ES™ instrument (Ion Torrent™). Next, semi-conductor NGS was performed in an Ion PGM™ System (Ion Torrent™) using the 400-bp chemistry of the Ion PGM™ Sequencing 400 Kit (Ion Torrent™).

The Torrent Suite™ 4.6 software (Ion Torrent™) was used for signal processing and basecalling. The generated FASTQ file containing the trimmed NGS reads was analyzed with the Alternative Splicing Detection Tool (ASDT) [39], an in-house-developed bioinformatic tool that facilitated the discovery of novel splice junctions and cryptic exons located in the intronic regions of the *DDC* gene.

### 4.6. RNA Expression Analysis of DDC in the cDNA Pools

Following the NGS analysis, the initial cDNAs with the same tissue origin were mixed in equal volumes, and thus 19 cDNA pools were generated. In each of the cDNA pools, the expression of *DDC* was examined using a PCR assay. Based on the existence of two distinct 5′ UTRs, *DDC* splice variants were divided into two subgroups. Therefore, a specific pair of primers for each of the two subgroups was used in the first-round PCR. All reactions were carried out at a final volume of 25 μL, while the reagents and the respective volumes were the same as in the previously mentioned first-round PCR assays (see Section 4.4), except for the concentration of KAPA Taq DNA polymerase, which was 0.5 units this time. The cycling conditions were the following: a denaturation step at 95 °C for 3 min, followed by 30 cycles of 95 °C for 30 s, 60 °C for 30 s, 72 °C for 2 min, and a final extension step at 72 °C for 2 min.

Next, each PCR product was diluted 100-fold in nuclease-free H_2_O. The diluted PCR products constituted the templates for the subsequent semi-nested PCR assays. The forward primer of the semi-nested PCR annealed to an inner site of exon two, while the reverse was the same primer used in the first PCR. The reaction was performed in 25 μL reaction mixtures consisting of the same ingredients as those listed above (see Section 4.4), in addition to KAPA Taq DNA Polymerase concentration (Kapa Biosystems Inc.), which equaled 0.5 units this time. The cycling protocol was the following: an initial denaturation step at 95 °C for 3 min, followed by 35 cycles of 95 °C for 30 s, 60 °C for 30 s, 72 °C for 2 min, and a final extension step at 72 °C for 2 min. All PCR amplifications were conducted in a Veriti 96-Well Fast Thermal Cycler (Applied Biosystems™). We focused on *DDC* expression in the 17 cDNA pools of cancer origin. In 11 of these pools, clear expression of *DDC* was observed under the conditions of the PCR assay. Therefore, the following experimental workflow was conducted in these 11 cDNA pools.

### 4.7. Determination of the Full Exon Structure of Each Novel Transcript with Nested PCR

The succeeding experimental procedure intended to identify the exon structure of the coding sequence of *DDC* transcripts, including the novel exons that ensued from the analysis of NGS findings, in the aforementioned 11 pools. In the present expression analysis, the novel exons deriving from our previous research study [20] were also included and the existence of splice variants encompassing these exons was investigated as well.

Two distinct nested PCR assays were required for the determination of the whole exon structure of each novel transcript; one started from exon 2 up to each novel exon, while the other one started from each novel exon up to exon 14. The aforementioned 100-fold diluted PCR products (Section 4.8) were used as templates for these reactions. The reaction mixture comprised the same reagents as in the first-round PCR (Section 4.8). The thermal protocol is analyzed below: an initial denaturation step at 95 °C for 3 min, followed by 35 cycles of 95 °C for 30 s, 60 °C for 30 s, 72 °C for 1 min or 1.5 min, depending on the size of the expected amplicon, and a final extension step at 72 °C for 1 min. All PCRs were conducted in a Veriti 96-Well Fast Thermal Cycler (Applied Biosystems™).

### 4.8. Expression Analysis of the DDC Novel Transcripts in the Seven CRC Cell Lines

Based on the findings from the expression analysis in the cDNA pools, the transcripts containing the novel exons that appeared to be more abundant in the CRC cDNA pool were further analyzed in each of the 7 CRC cell lines (COLO-205, Caco-2, HT-29, HCT-116, RKO, SW-620, DLD-1) individually. Firstly, a first-round PCR was performed, using a forward primer annealing to exon 2 and a reverse primer annealing to exon 14. The final volumes of the reaction mixture and its reagents were the same as the aforementioned ones (Section 4.8). The thermal protocol was the following: an initial denaturation step at 95 °C for 3 min, followed by 35 cycles of 95 °C for 30 s, 58 °C for 30 s, 72 °C for 2 min, and a final extension step at 72 °C for 2 min.

The first-round PCR products were diluted 50-fold and used as templates for the following semi-nested or nested PCR. As previously described, two PCR assays are required for the determination of the full exon structure of each splice variant. For the first PCR, the forward primer annealed to exon 2, to the same (or an inner) site as the one used in the first-round PCR, while the reverse primer was exon-specific for each one of the novel exons. The choice of the forward primer depended on the T_m_ of the reverse primer, so that the T_a_ is the optimum one. For the second PCR, the forward primer was specific for each novel exon and the reverse primer annealed to an inner site of exon 14 rather than the one used in the first-round PCR. The final volumes of the reaction mixture and its reagents were the same as the aforementioned ones (Section 4.8). The thermal protocol was the following: an initial denaturation step at 95 °C for 3 min, followed by 35 cycles of 95 °C for 30 s, 60 °C for 30 s, 72 °C for 1 min or 1.5 min, depending on the predicted product size, and a final extension step at 72 °C for 1 min. All PCR assays were conducted in a MiniAmp Thermal Cycler (Applied Biosystems™).

### 4.9. Agarose Gel Electrophoresis and Sanger Sequencing

All the aforementioned nested and semi-nested PCR products were electrophoresed on agarose gels and visualized under UV light by staining with Ethidium Bromide solution 1% (PanReac AppliChem ITW Reagents, Chicago, IL, USA). Each gel had a 1.5% agarose content. The pattern of the bands in each group of PCR products was evaluated and the predicted product size was calculated. The bands in the predicted product size (Appendix A) were properly excised from the gel and a purification process followed with the use of a Gel and PCR Clean-Up kit (Macherey-Nagel GmbH and Co. KG). Each of the purified products was quantitatively assessed with a high-sensitivity double-strand DNA kit in a Qubit^®^ 2.0 Fluorometer (Invitrogen™), then subjected to Sanger sequencing for amplicon sequence verification. The forward and reverse primers of each nested or semi-nested PCR were used as initiator sequences for the Sanger sequencing.

### 4.10. Construction of Structure Models and In Silico Analysis of the DDC Novel Protein Isoforms

Following the query of ORFs in the nucleotide sequences of the novel transcripts in CRC, the amino acid sequences of the new DDC protein isoforms were deduced. Next, 3D structure models were constructed by the I-TASSER server (https://zhanglab.ccmb.med.umich.edu/I-TASSER/) using multiple threading alignments and structural simulations [59,60], and the function of these proteins was predicted using COFACTOR (https://zhanglab.ccmb.med.umich.edu/COFACTOR/) [61] and COACH (https://zhanglab.ccmb.med.umich.edu/COACH/) [62,63]. An estimation of the accuracy of each prediction was made based on the confidence score of the modeling. In silico translational analysis to the deduced protein isoforms followed. For this purpose, the online tools from the ExPASy website (https://www.expasy.org/) were used [64]. More specifically, the molecular weight and the isoelectric points of these protein isoforms were calculated. The positions of post-translational modifications, including phosphorylation [NetPhos 3.1 Server (http://www.cbs.dtu.dk/services/NetPhos/)] and glycosylation [NetCGlyc 1.0 Server (http://www.cbs.dtu.dk/services/NetCGlyc/), NetNGlyc 1.0 Server (http://www.cbs.dtu.dk/services/NetNGlyc/), NetOGlyc 4.0 Server (http://www.cbs.dtu.dk/services/NetOGlyc/)], were also investigated. Additionally, the existence of palmitoylation sites was examined with the use of CSS-Palm 1.0 (http://csspalm.biocuckoo.org/) [65].

### 4.11. Pre-Amplification and Real-Time qPCR in Tissue Samples and CRC Cell Lines

Based on the results from the expression analysis in the cDNA pools and in cell lines, the novel exons with abundant expression in CRC cell lines were chosen for further quantification via a real-time qPCR assay in the 149 human tissue samples and in the 7 CRC cell lines. This assay was based on SYBR Green Chemistry and was performed in an ABI 7500 FAST Real-Time PCR System (Applied Biosystems™).

Due to the lower expression of these *DDC* novel exons compared to the already annotated ones, a pre-amplification PCR-based step was required prior to the real-time qPCR assay. Exon-specific pairs of primers were designed in the neighboring exons of each novel exon (Appendix A). The cDNA from each of the seven CRC cell lines and from the 149 tissue samples was used as a template for the first-round PCR. Based on existing literature, *HPRT1* was used as a reference gene [66] and was subjected to a pre-amplification PCR assay in all of the samples and cell lines. The reagents and their respective volumes are the same as in the aforementioned first-round PCR assay (Section 4.8). The thermal protocol is the following: a denaturation step at 95 °C for 3 min, followed by 25 cycles of 95 °C for 30 s, 58 °C for 30 s, 72 °C for 1 min, and a final extension step at 72 °C for 1 min. Each of the first-round PCR products was then diluted 1:50 and used as a template for the subsequent real-time qPCR assay. However, the first-round PCR products of the pre-amplification of *HPRT1* were diluted to 1:500 due to the higher expression of *HPRT1* compared to the *DDC* novel exon.

Follow this, real-time qPCR was conducted. The reaction mixture comprised 5 μL of KAPA SYBR FAST qPCR Master Mix (2X) Universal (it contains MgCl_2_ at a final concentration of 2.5 mM), supplemented with Low ROX (1X) as a passive reference dye, 1 μL forward primer and 1 μL reverse primer (final concentration of each primer: 200 nM), 2.5 μL DEPC-treated H_2_O and 0.5 μL first PCR product. For the quantification of each novel exon, an exon-specific pair of primers was designed to anneal to the sequence of this novel exon (Appendix A). The product size of each amplicon is displayed in Appendix A.

Duplicates of every sample (and triplicates of the cell lines) were used in each qPCR assay in order to assess the reproducibility of the obtained data. The cycling steps were the following: (1) 3 min at 95 °C for denaturation; (2) 40 cycles consisting of two steps: 3 s at 95 °C so that the PCR products became denatured, then 30 s at 60 °C for primer annealing and extension; (3) generation of a melting curve, aiming to distinguish the main product from the primer dimers and other non-specific products; the latter usually have a lower T_m_ (<75 °C) than the main product. The melt curves of the produced amplicons are shown in Appendix A.

### 4.12. Calculations and Validation of the Comparative C_T_ (2^−∆∆CT^) Method for DDC Novel Exon Quantification

After the completion of the real-time qPCR experiment, the *DDC* transcripts containing one of the selected novel exons were quantified using the comparative C_T_ (2^−∆∆C^_T_) method. The diluted PCR product from the first-round PCR of the CRC cDNA pool was used as a calibrator in each real-time qPCR assay. For the normalization of *DDC* transcripts that comprise one of these novel exons, the mean expression of the housekeeping gene was used as a reference. The relative *DDC* novel exon expression of each sample or cell line resulted from the division of the normalized *DDC* novel exon expression in the target sample by the respective ratio calculated for the calibrator, and was measured in RQU. In samples with undetectable expression, a conventional expression value of 0.001 RQU was assigned; this value was equal to the lowest quantifiable expression level.

This method is based on the hypothesis that the PCR amplification efficiency between the reference and target genes is similar and at the optimum level for both genes. A validation experiment (construction of standard curve) was performed so that the prerequisites of the comparative method could be checked (Appendix A). The C_T_ values of *DDC* novel exons and the reference gene were measured in dilution series of the first PCR product of the cDNA pool. This pool was the calibrator in this experiment so that PCR samples resulting from different runs became comparable. The qPCR efficiency was calculated based on the following formula: $E=-1+10^{\left(-1/a\right)}$, where α represents the slope of the corresponding amplification plot. This experiment was conducted for all of the selected *DDC* novel exons and *HPRT1*.

### 4.13. Biostatistical Analysis

Extensive biostatistical analysis of the results was performed. More specifically, the Wilcoxon signed-rank test was implemented, so that the expression levels of *DDC* novel exons were compared among the 40 paired samples. A potential association between each *DDC* novel exon expression status and the clinicopathological characteristics was investigated via Jonckheere–Terpstra test and Mann–Whitney test. Prior to further analysis, an optimal cut-off point was determined for each of the *DDC* novel exons using the X-tile software, so that the patients were divided into *DDC* novel exon-positive and -negative groups.

Survival analysis was performed via the construction of Kaplan–Meier DFS and OS curves, and the differences that emerged were assessed via the log-rank (Mantel–Cox) test. Additionally, the assessment of the association between the prognostic markers and the relative risks for disease recurrence and patient death was performed through the development of proportional hazard Cox regression models. All parameters designated as statistically significant from the univariate Cox regression analysis were included in the multivariate Cox regression models. Finally, patients were stratified into groups with distinct prognoses based on clinical factors with already established prognostic value. Stratified Kaplan–Meier curves were constructed for the evaluation of the prognostic potential of the *DDC* novel exon expression status regarding DFS and OS in the aforementioned groups of patients. Bootstrapping was conducted for the Cox regression models using 1000 samples, and bias-corrected and accelerated (BCa) 95% confidence intervals (CI) of each estimated hazard ratio (HR) were estimated.

Outcome with *p* values of less than 0.050 were defined as statistically significant.

## 5. Conclusions

Recently published NGS data from our research group prove the existence of 6 *DDC* novel exons in human cancer cell lines. The present study led to the discovery of 3 additional novel exons and 20 novel transcripts of *DDC* via an NGS analysis; 4 of these novel exons and 8 of these novel transcripts are expressed in CRC cell lines. In silico analysis demonstrated the potential existence of 6 protein isoforms, 2 of which were expressed in CRC cell lines. Considering the pivotal involvement of catecholamines and biogenic amines in general in CRC initiation and progression, as well as the critical role of DDC in their biosynthetic pathways, the discovery of additional DDC protein isoforms sets new questions regarding the function of this enzyme. Additionally, the presence of novel exons with potential prognostic value in patients with tumors in TNM stages II and III is quite encouraging, due to the urgent need for a better stratification system that is capable of predicting the disease progression of patients in these stages. The potential participation of DDC and its products in neurological, immunological, and cancer responses indicates this molecule as being critical for cell viability and normal cell function, necessitating further investigation.

## Figures and Tables

**Figure 1 ijms-21-08568-f001:**
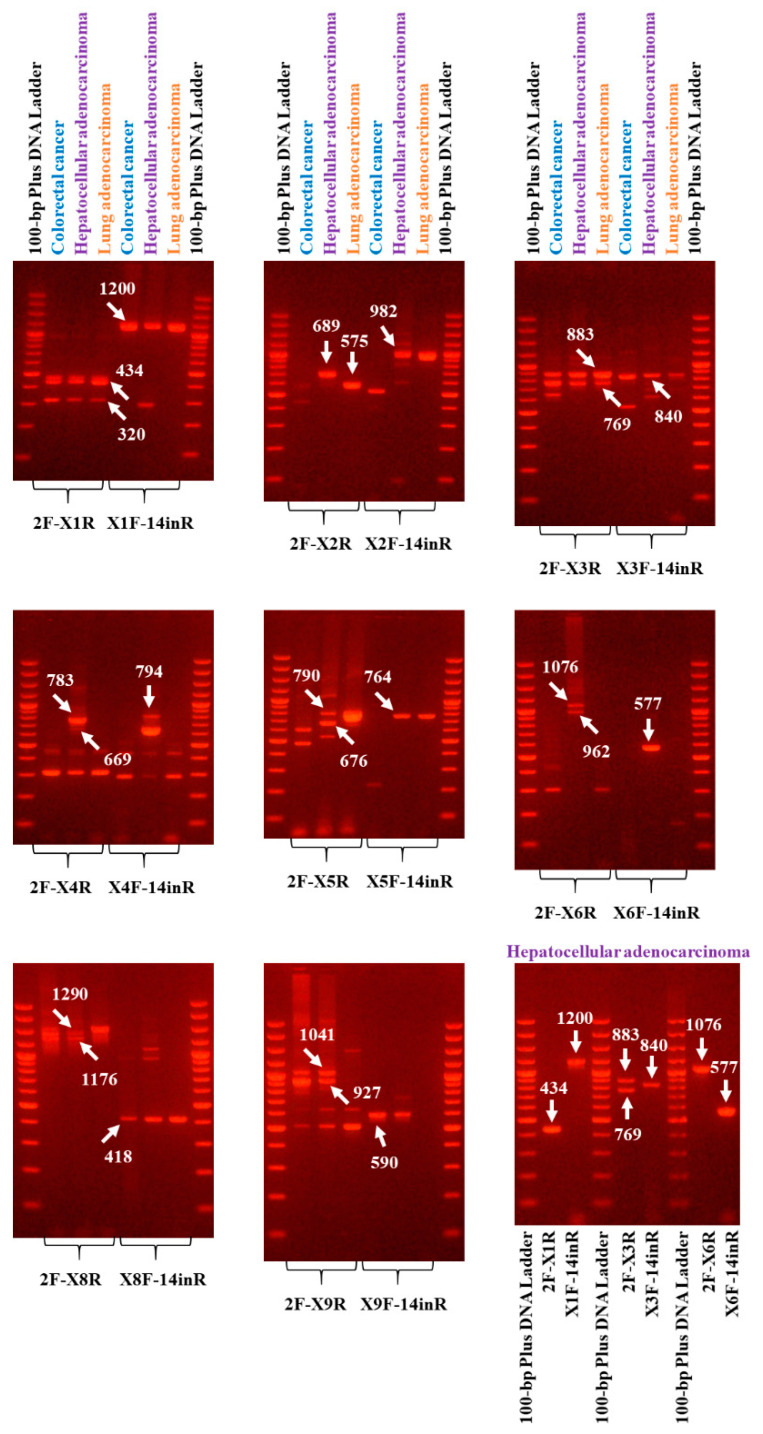
Expression analysis in the 3 cDNA pools derived from colorectal, hepatocellular, and lung cancerous tissues, in which the novel transcripts were more abundant. More specifically, the transcripts with L1 as the first exon containing one of the novel exons (X1 to X9) are shown in the first 8 photos. The transcripts starting with exon N1 are shown in the last photo (bottom right). The pair of primers that was used for the generation of each transcript and the expected product size (s) are written on the panel. The arrows point at each predicted product.

**Figure 2 ijms-21-08568-f002:**
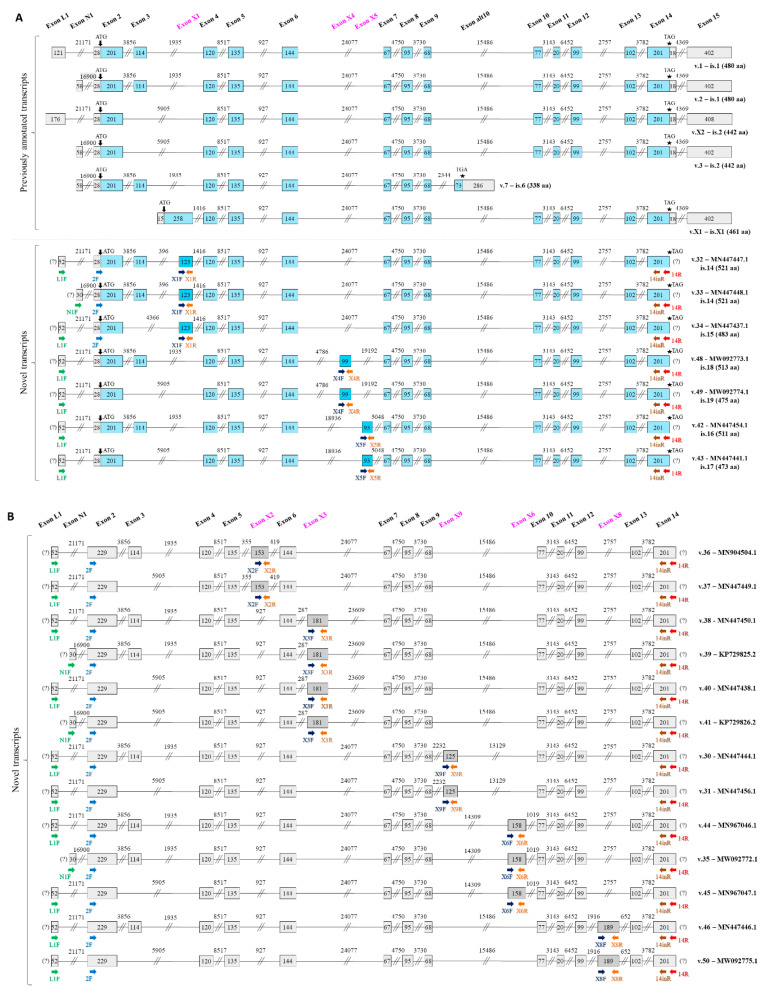
Depiction of the exon structure of the 20 *DDC* novel transcripts. The names of the novel exons are shown in pink. Exons are presented as boxes and introns as lines, while numbers inside boxes and above lines indicate the length of each exon or intron (in nucleotides), respectively. In the first-round PCR, either of the forward (green color) primers and the common reverse primer (red color) were utilized together. The forward primer (light blue color) in the nested PCR was used along with an exon-specific reverse primer (orange color) for the amplification of the left part of each transcript; the exon-specific forward primer (dark blue color) in the nested PCR was used along with a common reverse primer (brown color) for the amplification of the right part of each transcript. (**A**) Previously annotated *DDC* transcripts and the 7 *DDC* novel transcripts with an open reading frame (ORF) are depicted, accompanied by their accession numbers. Blue and grey boxes indicate coding and non-coding exons, respectively, while novel exons are colored with a more intense blue shade. Arrows (↓) indicate the position of the ATG codon, pentagrams (★) show the position of the stop codon, and question marks (?) represent an undetermined untranslated region (UTR). (**B**) *DDC* novel transcripts without an ORF, accompanied by their accession numbers. Novel exons are colored dark gray.

**Figure 3 ijms-21-08568-f003:**
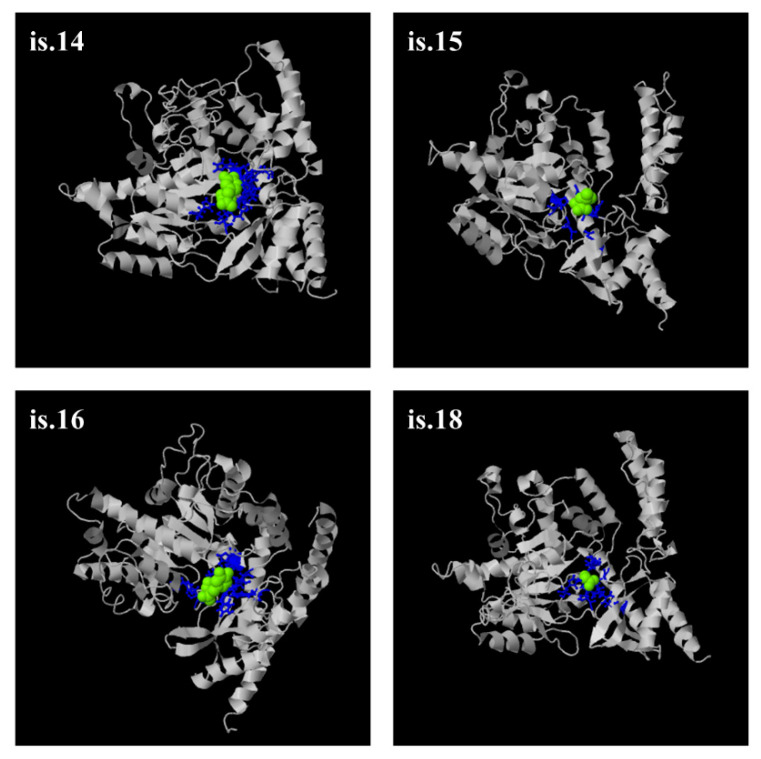
Predicted ligands bound to 4 of the deduced DDC isoforms (is.14, is.15, is.16, and is.18), illustrated with the use of the I-TASSER, COFACTOR (https://zhanglab.ccmb.med.umich.edu/COFACTOR/), and COACH (https://zhanglab.ccmb.med.umich.edu/COACH/) servers. Each ligand is depicted with a green-yellow sphere. Binding aa residues are shown by blue balls and sticks. Besides pyridoxal-5-phosphate (PLP), is.14 and is.16 bind carbidopa, a well-known inhibitor of DDC function. Is.15 binds the amino acid leucine and is.18 binds the amino acid glycine. All these predictions have relatively high confidence scores.

**Figure 4 ijms-21-08568-f004:**
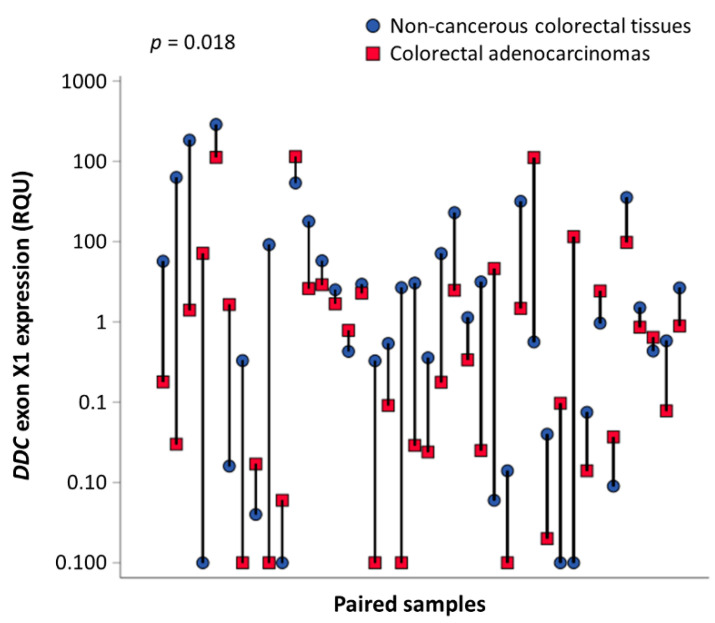
Comparison of *DDC* exon X1 expression levels among 40 CRC tissue samples and their adjacent non-cancerous counterparts. *DDC* exon X1 expression is downregulated in colorectal tumors as compared to non-cancerous paired tissues (*p* = 0.018). The *p* value was calculated using the Wilcoxon signed-rank test.

**Figure 5 ijms-21-08568-f005:**
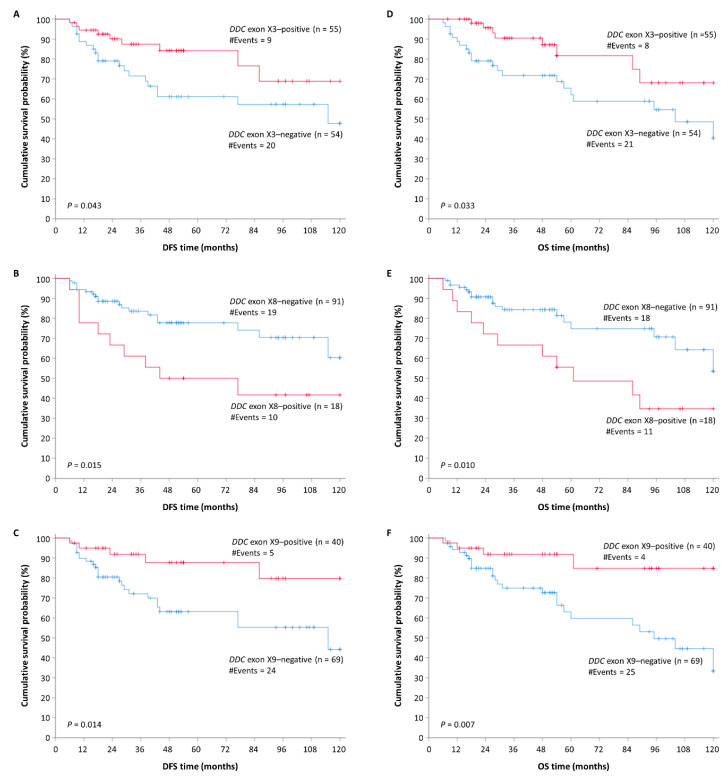
Kaplan–Meier survival curves for the disease-free survival (DFS) and overall survival (OS) times of CRC patients. (**A**) Positive *DDC* exon X3 expression status is a potential favorable prognostic biomarker of DFS in CRC (*p* = 0.043). (**B**) *DDC* X8-positive patients have a shorter DFS time period than the negative ones (*p* = 0.015). (**C**) Positive *DDC* exon X9 expression status leads to longer CRC patient DFS time (*p* = 0.014). (**D**) Patients with *DDC* exon X3-positive tumors have longer OS period than those with *DDC* exon X3-negative tumors (*p* = 0.033). (**E**) Positive *DDC* exon X8 expression status is a potential unfavorable prognostic biomarker of OS in CRC (*p* = 0.010). (**F**) *DDC* exon X9-positive patients are characterized by a longer OS time period than the negative ones (*p* = 0.007).

**Figure 6 ijms-21-08568-f006:**
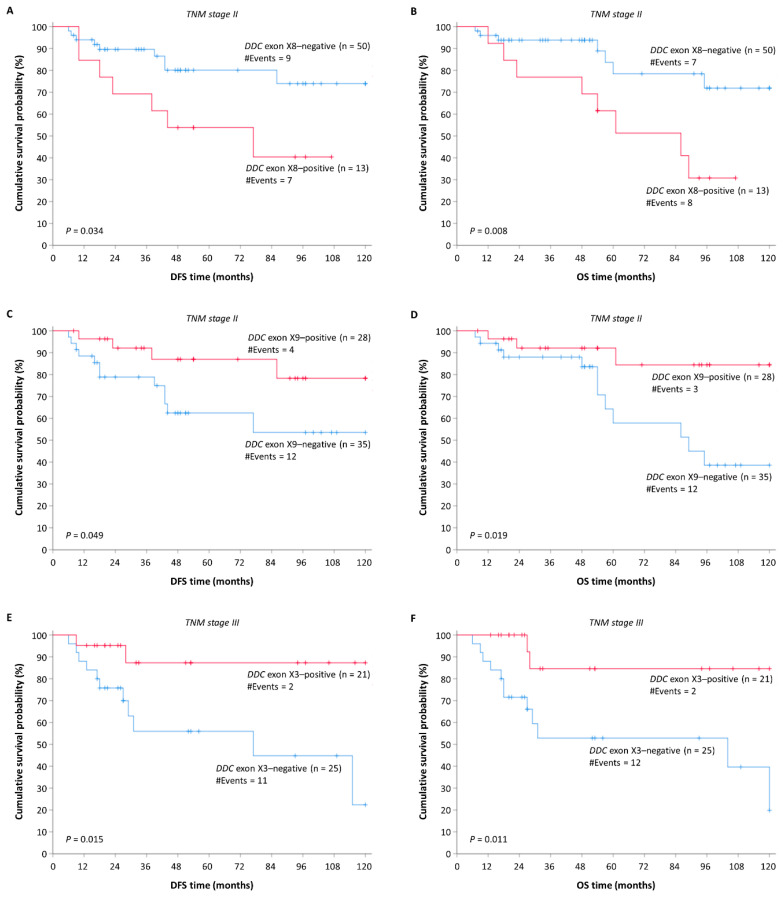
Stratified Kaplan–Meier survival curves of groups of patients with substantially different prognoses according to TNM stage. Patients with *DDC* exon X8-positive tumors in TNM stage II have (**A**) a shorter DFS and (**B**) OS time period compared to those with *DDC* exon X8-negative tumors. Patients with *DDC* exon X9-positive tumors in TNM stage II have (**C**) a higher DFS and (**D**) OS time period compared to those with *DDC* exon X9-negative tumors. Patients with *DDC* exon X3-positive tumors in TNM stage III have (**E**) a higher DFS and (**F**) OS time period compared to those with *DDC* exon X3-negative tumors.

**Figure 7 ijms-21-08568-f007:**
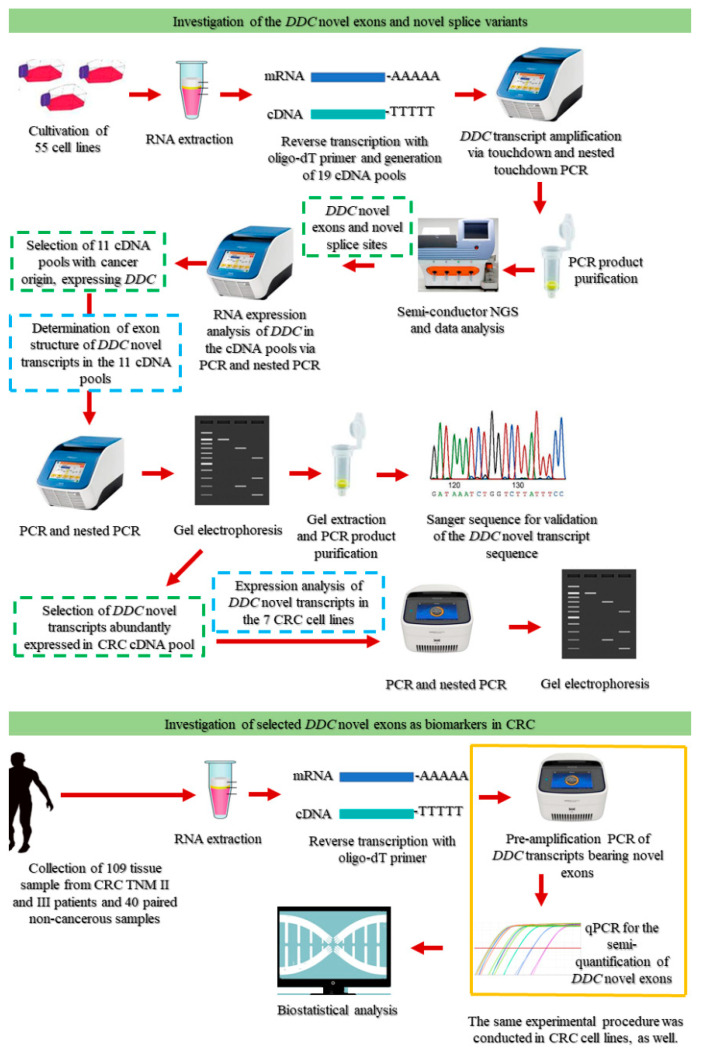
Overview of the experimental workflow.

**Table 1 ijms-21-08568-t001:** Properties of the deduced DDC protein isoforms and putative post-translational modification sites residing in their novel peptides, based on bioinformatical analysis.

DDC Isoform	Length (aa ^1^ Residues)	pI ^2^	M_r_ ^3^ (kDa)	Post-Translational Modification	Kinase	Site	Position
is.14	521	7.22	58.0	Phosphorylation	PKA ^4^	Ser	113
					unspecified	Tyr	144
				S-palmitoylation		Cys	107
						Cys	108
is.15	483	7.55	53.8	Phosphorylation	PKA^4^	Ser	75
					unspecified	Tyr	106
				S-palmitoylation		Cys	69
						Cys	70
is.16	511	6.63	57.3	Phosphorylation	PKC ^5^	Thr	245
					CDK1 ^6^	Ser	257
					GSK3 ^7^	Ser	257
					CSNK1^8^	Ser	260
					PKC ^5^	Ser	260
					PKC ^5^	Ser	264
is.17	473	6.71	53.0	Phosphorylation	PKC ^5^	Thr	207
					CDK1 ^6^	Ser	219
					GSK3 ^7^	Ser	219
					CSNK1 ^8^	Ser	222
					PKC ^5^	Ser	222
					PKC ^5^	Ser	226
is.18	513	7.24	57.9	Phosphorylation	EGFR ^9^	Tyr	244
					PKA ^4^	Ser	257
is.19	475	7.59	53.6	Phosphorylation	EGFR ^9^	Tyr	206
					PKA ^4^	Ser	219

Note: ^1^ amino acid; ^2^ isoelectric point; ^3^ molecular mass. ^4^ cyclic adenosine monophosphate (cAMP)-activated protein kinase; ^5^ protein kinase C; ^6^ cyclin-dependent kinase 1; ^7^ glycogen synthase kinase 3; ^8^ casein kinase 1; ^9^ epidermal growth factor receptor.

**Table 2 ijms-21-08568-t002:** Characterization of the 109 colorectal cancer (CRC) cases and expression of *DDC* novel exons in cancerous and non-cancerous colorectal tissue samples.

		Number of Patients (%)			
**Gender**					
Male		62 (56.9%)			
Female		47 (43.1%)			
**Tumor site**					
Colon		72 (66.1%)			
Rectum		37 (33.9%)			
**Histological grade**					
I		8 (7.3%)			
II		84 (77.1%)			
III		17 (15.6%)			
**Venous invasion**					
Absent		86 (78.9%)			
Present		23 (21.1%)			
**Lymphatic invasion**					
Absent		92 (84.4%)			
Present		17 (15.6%)			
**T (tumor invasion)**					
T1		0 (0.0%)			
T2		3 (2.8%)			
T3		86 (78.9%)			
T4		20 (18.3%)			
**N (nodal status)**					
N0		63 (57.8%)			
N1		33 (30.3%)			
N2		13 (11.9%)			
**TNM stage ^1^**					
II		63 (57.8%)			
III		46 (42.2%)			
		Percentiles			
**Variable**	**Mean ± S.E. ^2^**	**Range**	**25th**	**50th (Median)**	**75th**
***DDC* exon X1 expression (RQU** **^3^)**					
in malignant tumors (*n* = 109)	144.74 ± 76.96	0.001–7538.2	0.025	0.31	2.95
in non-cancerous tissues (*n* = 40)	18.72 ± 8.51	0.001–288.0	0.049	1.05	6.78
***DDC* exon X3 expression (RQU** **^3^)**					
in malignant tumors (*n* = 109)	26.65 ± 21.72	0.001–2368.9	0.001	0.32	3.18
in non-cancerous tissues (*n* = 40)	26.58 ± 21.99	0.001–879.2	0.012	0.47	2.93
***DDC* exon X8 expression (RQU** **^3^)**					
in malignant tumors (*n* = 109)	11.22 ± 5.6	0.001–552.6	0.001	0.20	1.80
in non-cancerous tissues (*n* = 40)	56.69 ± 39.38	0.001–1499.2	0.001	0.001	0.098
***DDC* exon X9 expression (RQU** **^3^)**					
in malignant tumors (*n* = 109)	8.48 ± 7.52	0.001–820.3	0.001	0.017	0.53
in non-cancerous tissues (*n* = 40)	24.27 ± 16.63	0.001–572.1	0.001	0.001	0.86
**Patient age (years)**	66 ± 1.2	35–93	58	68	75
**Tumor size (cm)**	4.7 ± 0.2	2–12	3.5	4.5	5.5

^1^ Tumor, node, and metastasis classification. ^2^ Standard error. ^3^ Relative quantification unit.

**Table 3 ijms-21-08568-t003:** Expression status of *DDC* novel exons and DFS of CRC patients.

	Univariate Analysis (*n* = 109)			Multivariate Analysis ^1^ (*n* = 109)		
Covariate	HR ^2^	BCa 95% Bootstrap CI ^3^	Bootstrap *p* Value ^4^	HR ^2^	BCa 95% Bootstrap CI ^3^	Bootstrap *p* Value ^4^
***DDC* exon X1 expression status**						
Negative (*n* = 54)	1.00					
Positive (*n* = 55)	0.67	0.31–1.31	0.27			
***DDC* exon X3 expression status**						
Negative (*n* = 54)	1.00			1.00		
Positive (*n* = 55)	0.45	0.20–0.91	0.033	0.42	0.14–0.82	*0.037*
***DDC* exon X8 expression status**						
Negative (*n* = 91)	1.00			1.00		
Positive (*n* = 18)	2.51	1.00–5.69	0.010	3.17	1.05–14.93	*0.011*
***DDC* exon X9 expression status**						
Negative (*n* = 69)	1.00			1.00		
Positive (*n* = 40)	0.32	0.097–0.64	0.015	0.32	0.10–0.60	*0.009*
**Tumor site**						
Colon (*n* = 72)	1.00			1.00		
Rectum (*n* = 37)	2.52	1.16–6.12	0.010	2.16	0.91–5.86	*0.034*
**Histological grade**						
I (*n* = 8)	1.00					
II (*n* = 84)	1.00	0.34–2.7 × 10^4^	0.99			
III (*n* = 17)	2.90	0.69–8.4 × 10^4^	0.077			
**Venous invasion**						
Absent (*n* = 86)	1.00					
Present (*n* = 23)	1.68	0.63–3.44	0.22			
**Lymphatic invasion**						
Absent (*n* = 92)	1.00					
Present (*n* = 17)	1.77	0.69–3.72	0.15			
**TNM stage**						
II (*n* = 63)	1.00					
III (*n* = 46)	1.27	0.58–2.58	0.54			

^1^ Multivariate models regarding DFS were adjusted for the tumor site. ^2^ Hazard ratio, estimated from proportional hazard Cox regression models. ^3^ Bias-corrected and accelerated 95% confidence interval of the estimated HR. ^4^ Statistically significant *p* values are shown in italics.

**Table 4 ijms-21-08568-t004:** Expression status of *DDC* novel exons and OS of CRC patients.

	Univariate Analysis (*n* = 109)			Multivariate Analysis ^1^ (*n* = 109)		
Covariate	HR ^2^	BCa 95% Bootstrap CI ^3^	Bootstrap *p* Value ^4^	HR ^2^	BCa 95% Bootstrap CI ^3^	Bootstrap *p* Value ^4^
***DDC* exon X1 expression status**						
Negative (*n* = 54)	1.00					
Positive (*n* = 55)	0.76	0.33–1.57	0.43			
***DDC* exon X3 expression status**						
Negative (*n* = 54)	1.00			1.00		
Positive (*n* = 55)	0.42	0.15–0.87	0.031	0.31	0.095–0.61	*0.016*
***DDC* exon X8 expression status**						
Negative (*n* = 91)	1.00			1.00		
Positive (*n* = 18)	2.60	1.16–5.84	0.006	3.63	1.18–32.25	*0.011*
***DDC* exon X9 expression status**						
Negative (*n* = 69)	1.00			1.00		
Positive (*n* = 40)	0.26	0.061–0.55	0.012	0.23	0.078–0.42	*0.004*
**Tumor site**						
Colon (*n* = 72)	1.00			1.00		
Rectum (*n* = 37)	2.43	1.21–4.77	0.009	2.06	0.92–6.26	0.060
**Histological grade**						
I (*n* = 8)	1.00					
II (*n* = 84)	0.97	0.30–2.5 × 10^4^	0.95			
III (*n* = 17)	2.80	0.56–9.5 × 10^4^	0.095			
**Venous invasion**						
Absent (*n* = 86)	1.00					
Present (*n* = 23)	1.81	0.60–4.64	0.18			
**Lymphatic invasion**						
Absent (*n* = 92)	1.00					
Present (*n* = 17)	1.76	0.74–3.39	0.14			
**TNM stage**						
II (*n* = 63)	1.00					
III (*n* = 46)	1.56	0.74–3.47	0.24			

^1^ Multivariable models regarding OS were adjusted for the tumor site. ^2^ Hazard ratio, estimated from proportional hazard Cox regression models. ^3^ Bias-corrected and accelerated 95% confidence interval of the estimated HR. ^4^ Statistically significant *p* values are shown in italics.

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
