# Peer review of "Revised Exon Structure of l-DOPA Decarboxylase (DDC) Reveals Novel Splice Variants Associated with Colorectal Cancer Progression"

_ijms, 2020, doi:10.3390/ijms21228568_

Round 1
Reviewer 1 Report
This original research article by Artemaki et al. is very interesting and innovative, at the same time. It addresses the need for additional prognostic biomarkers in colorectal cancer (CRC), particularly in patients of TNM stages II and III, whose prognosis is not substantially different. More interestingly, these molecular biomarkers are novel RNA molecules, which have been identified using a sound methodological approach: the combination of high-throughput sequencing with nested PCR and Sanger sequencing. Some of these mRNAs probably encode isoenzymes of this Decarboxylase, as shown by bioinformatical analysis. Moreover, the biostatistical analysis comprises the application of the appropriate statistical tests, while bootstrapping in the survival analysis renders the results even more solid. I have only a few minor suggestions to do:
- The authors should consider adding a Figure that clearly presents the methodology of this research study. This would help the readers to better understand their experimental approach.
- In the Figures showing 3D structures of the new predicted isoforms of DDC, the 3D structures of main DDC protein isoform(s) should be added.
- Do the main DDC protein isoform(s) have different substrates, compared to the newly discovered ones? This issue should be clearly discussed.
- Since it is the very first time that these DDC exons / groups of transcripts are being quantified, it would make sense perhaps to show qPCR standard curves and melting curves of the produced amplicons. This suggested Figure could be either a main one or a Supplemental one.
- Did the authors consider adding TNM stage as a covariate in the multivariable Cox regression?
- I observed that the lowest expression values of all exons in Table 2 are the same. This could be attributed only to a convention that it is currently not clearly stated; for instance, a conventional expression value for undetected expression in one (or more) sample(s). If so, this should be clearly stated in the text.
- It is not quite clear whether the other discovered exons are present in DDC transcripts expressed in colorectal cell lines and/or tissue samples, or not. This information should be added.
- Extensive English correction is needed; currently, the text contains several grammar and syntax errors. Moreover, there are a few typing errors that need to be corrected.
- A couple of important references have been omitted, in my opinion. The authors are advised to add some information about DDC expression in human tissues, particularly in human malignancies (both solid tumors and hematological malignancies).
Author Response
Reviewer #1 (Comments to the Author):
- The authors should consider adding a Figure that clearly presents the methodology of this research study. This would help the readers to better understand their experimental approach.
As suggested by the Reviewer, we added a Figure which clearly presents the methodology of this research study:
- Page 20, lines 552-553 (Materials and methods): An overview of the experimental workflow that was followed in the present study is depicted in Figure 7.
- In the Figures showing 3D structures of the new predicted isoforms of DDC, the 3D structures of main DDC protein isoform(s) should be added.
Regarding the Reviewer’s comment, we added the structures of the 3 main DDC protein isoforms in Figure S2 with the following modified Figure legend: “Figure S2. Predicted 3D structures for the main and the deduced DDC protein isoforms. For each protein, only the 3D structure with the highest confidence score is demonstrated. α-helixes and β-sheets are drawn with pink and yellow color, respectively, while coils are depicted as white strings.”
- Page 3, lines 131-132 (Results): […] exists in the alt-DDC transcript (accession number: NM_001242890.2, encodes DDC is.6), …
- Page 6, lines 197-199 (Results): Most of the deduced protein isoforms are larger compared to the main DDC isoform (is.1) of 480 aa residues, to DDC is.2 of 442 aa residues (the internal peptide encoded by exon 3 is absent), and to DDC is.6 of 338 aa residues.
- Page 6, lines 203-205 (Results): The 3D protein structures with the highest confidence score were built by the I-TASSER server and they demonstrate that all 6 novel protein isoforms contain a distinct combination of α-helixes, β-sheets and coils, compared to DDC is.1, is.2, and is.6 (Figure S2).
- Do the main DDC protein isoform(s) have different substrates, compared to the newly discovered ones? This issue should be clearly discussed.
Taking into consideration the Reviewer’s comment, we decided to further discuss this issue:
- Page 6, lines 208-209 (Results): The ligands carbidopa and glycine bind also to DDC is.6 and is.1, respectively.
- Page 18, lines 440-446 (Discussion): For instance, the well-known DDC inhibitor [40] carbidopa is predicted to bind with high confidence score only to is.14 and is.16, a finding which raises questions regarding the regulation of DDC activity. A previously unaccounted interaction of these DDC protein isoforms with the amino acids leucine and glycine also arose from this investigation. The interaction with other ligands besides PLP may lead to conformational changes of DDC isoforms and, hence, to their binding to different substrates. Therefore, the hypothesis regarding the different substrates and functions of the deduced protein isoforms is supported.
- Since it is the very first time that these DDC exons / groups of transcripts are being quantified, it would make sense perhaps to show qPCR standard curves and melting curves of the produced amplicons. This suggested Figure could be either a main one or a Supplemental one.
As suggested by the Reviewer, we included two additional figures to show the melt curves and the standard curves of the produced amplicons: Figure S6 and Figure S7, respectively.
- Page 25, lines 749-750 (Materials and methods): The melt curves of the produced amplicons are shown in Figure S6.
- Page 26, lines 763-765 (Materials and methods): A validation experiment (construction of standard curves) was performed so that the prerequisites of the comparative method be checked (Figure S7).
Their figure legends are the following:
“Figure S6. Melt curves of the qPCR products of the novel DDC exons. (a) DDC exon X1, (b) DDC exon X3, (c) DDC exon X8, and (d) DDC exon X9. Figure S7. Standard curves built by plotting the threshold cycle (CT) versus the dilution. For the standard curves, PCR products generated from pre-amplification of CRC cDNA pool were used. (a) Standard curve for DDC exon X1; PCR product of CRC cDNA pool was 4-fold serially diluted. (b) Standard curve for DDC exon X3; PCR product of CRC cDNA pool was 4-fold serially diluted. (c) Standard curve for DDC exon X8; PCR product of CRC cDNA pool was 2-fold serially diluted. (d) Standard curve for DDC exon X9; PCR product of CRC cDNA pool was 4-fold serially diluted.”
- Did the authors consider adding TNM stage as a covariate in the multivariable Cox regression?
The cohort of the present study included only TNM stage II and III CRC patients. We have investigated the effect of TNM stage in the univariate Cox regression regarding disease free survival (DFS) and overall survival (OS). However, it did not prove to be statistically significant, and therefore, we decided not to include it in the multivariable Cox regression analysis. This finding is in consistency with the inherent weakness of TNM staging to accurately predict the survival outcome of TNM II and III CRC patients.
- I observed that the lowest expression values of all exons in Table 2 are the same. This could be attributed only to a convention that it is currently not clearly stated; for instance, a conventional expression value for undetected expression in one (or more) sample(s). If so, this should be clearly stated in the text.
We thank the Reviewer for this remark. We clarified this issue:
Page 26, lines 759-761 (Materials and methods): In samples with undetectable expression, a conventional expression value of 0.001 RQU was attributed; this value was equal to the lowest quantifiable expression.
- It is not quite clear whether the other discovered exons are present in DDC transcripts expressed in colorectal cell lines and/or tissue samples, or not. This information should be added.
Taking into consideration the Reviewer’s comment, we added the required information:
- Page 6, lines 176-180 (Results): The rest 5 novel exons were either detected in low abundance or not detected at all, with this particular experimental protocol, in the CRC cDNA pool. Based on this finding, only the aforementioned 4 novel exons were further investigated in the 7 CRC cell lines. However, the possibility that the rest 5 novel exons are expressed in specific CRC cell lines should not be excluded.
- Page 9, lines 242-243 (Results): Based on the abundance of DDC exons X1, X3, X8, and X9 in the CRC cDNA pool, the expression of these 4 novel exons was next quantified in human tissue samples.
- Extensive English correction is needed; currently, the text contains several grammar and syntax errors. Moreover, there are a few typing errors that need to be corrected.
According to the Reviewer’s suggestion, we corrected typing, grammatical, and syntax errors throughout the manuscript.
- A couple of important references have been omitted, in my opinion. The authors are advised to add some information about DDC expression in human tissues, particularly in human malignancies (both solid tumors and hematological malignancies).
We thank the Reviewer for this remark. We included the following references:
- Patsis, C.; Glyka, V.; Yiotakis, I.; Fragoulis, E.G.; Scorilas, A. l-DOPA Decarboxylase (DDC) Expression Status as a Novel Molecular Tumor Marker for Diagnostic and Prognostic Purposes in Laryngeal Cancer. Transl Oncol 2012, 5, 288-296, doi:10.1593/tlo.12223.
- Geomela, P.A.; Kontos, C.K.; Yiotakis, I.; Fragoulis, E.G.; Scorilas, A. L-DOPA decarboxylase mRNA expression is associated with tumor stage and size in head and neck squamous cell carcinoma: a retrospective cohort study. BMC Cancer 2012, 12, 484, doi:10.1186/1471-2407-12-484.
- Korbakis, D.; Fragoulis, E.G.; Scorilas, A. Quantification and study of the L-DOPA decarboxylase expression in gastric adenocarcinoma cells treated with chemotherapeutic substances. Anticancer Drugs 2013, 24, 291-299, doi:10.1097/CAD.0b013e32835db25a.
- Papadopoulos, E.I.; Petraki, C.; Gregorakis, A.; Chra, E.; Fragoulis, E.G.; Scorilas, A. L-DOPA decarboxylase mRNA levels provide high diagnostic accuracy and discrimination between clear cell and non-clear cell subtypes in renal cell carcinoma. Clin Biochem 2015, 48, 590-595, doi:10.1016/j.clinbiochem.2015.02.007.
- Avgeris, M.; Koutalellis, G.; Fragoulis, E.G.; Scorilas, A. Expression analysis and clinical utility of L-Dopa decarboxylase (DDC) in prostate cancer. Clin Biochem 2008, 41, 1140-1149, doi:10.1016/j.clinbiochem.2008.04.026.
The authors wish to thank the Reviewers for their constructive comments that led to the improvement of the current manuscript.

Reviewer 2 Report
In their manuscript titled “Revised exon structure of L-DOPA decarboxylase (DDC) reveals novel splice variants associated with colorectal cancer”, the authors analyzed alternative splicing of transcripts encoding the L-DOPA decarboxylase (DDC). Next generation sequencing was performed in 55 human cell lines and resulted in identification of 3 novel DDC exons, namely X2, X7, and X9. For the analyses presented in this study, the authors combined these three novel exons with six previously identified exons, namely X1, X3, X4, X5, X6, and X8 and identified 20 novel splice variants. 8 of these 20 splice variants were also detected in tissue samples of CRC patients. Furthermore, they found that specific splice variants, containing the exons X3, X8, and X9, have prognostic value for the disease-free and overall survival of these patients.
The results of this study are of interest to a broad readership, since the authors describe a new approach to identify potential biomarkers for colorectal cancer. Furthermore, the observation that the expression of exons X3 and X9 changes with the progression of TMN II to TMN III as well as the resulting clinical and functional implications add to the value of this study. The results are presented clearly, and the conclusions are supported by the data. For these reasons, I recommend publishing this manuscript. Nevertheless, I kindly ask the authors to address the minor points mentioned below.
Minor points:
- Introduction, line 116: What was the rationale for selecting colorectal carcinomas with the TMN stages II and III. Please add a statement explaining your choice in the manuscript.
- Results, Section 2.5, first paragraph: The authors present data showing that the expression of exon X1 is reduced in tumor tissue compared to normal tissue. Did you perform a ROC analysis for X1? What is the AUC value for exon X1 when comparing tumor tissue and normal tissue?
- Discussion, Line 401ff: The analyses presented in this manuscript were based on colorectal tumor tissue and paired normal colonic tissue. Have you detected the 8 novel DDC splice variants in serum or plasma samples of CRC patients? Please add a corresponding statement to the discussion of the manuscript.
- Discussion, Line 401ff: The authors used the TMN classification to stratify the patient collection. Is the expression of the exons X1, X3, X8, and X9 associated with specific subtypes of colorectal tumors, e.g. CIN, MIS, CIMP? Please add a corresponding statement to the discussion of the manuscript.
Author Response
Reviewer #2 (Comments to the Author):
- Introduction, line 116: What was the rationale for selecting colorectal carcinomas with the TMN stages II and III. Please add a statement explaining your choice in the manuscript.
We thank the Reviewer for this comment. We added the following information:
Page 3, lines 116-120 (Introduction): For this reason, they were selected for quantification of their expression levels, via an in-house developed qPCR assay, in 109 CRC samples from patients in TNM stage II and III and 40 adjacent normal ones. The present study focused on CRC TNM stage II and III patients, due to the inefficiency of the current TNM staging to predict the survival outcome of CRC patients classified in these subgroups.
- Results, Section 2.5, first paragraph: The authors present data showing that the expression of exon X1 is reduced in tumor tissue compared to normal tissue. Did you perform a ROC analysis for X1? What is the AUC value for exon X1 when comparing tumor tissue and normal tissue?
For the comparison of the expression levels between cancerous and non-cancerous state, only paired tissue samples were used. Thus, the Wilcoxon signed-rank test was implemented to compare the expression levels of DDC novel exons among the 40 paired samples.
In our opinion, the ROC analysis is not applicable in the present study, as the P value that accompanies ROC analysis is calculated using the Mann-Whitney U test, a non-parametric test that is used for comparisons between two independent samples (i.e. not two related samples).
- Discussion, Line 401ff: The analyses presented in this manuscript were based on colorectal tumor tissue and paired normal colonic tissue. Have you detected the 8 novel DDC splice variants in serum or plasma samples of CRC patients? Please add a corresponding statement to the discussion of the manuscript.
In response to the Reviewer’s remark, we added the following statement:
Page 19, lines 482-486 (Discussion): Prompted by this potential of splice variants, we investigated the value of the specific DDC novel exons as biomarkers, and specifically of those that were observed as more abundant from the expression analysis in the cDNA pools. Their utility as biomarkers was evaluated in CRC tissue samples; however, due to the emerging value of liquid biopsy and its daily use in clinical practice, it would be interesting to evaluate these DDC novel exons in serum or plasma of CRC patients.
- Discussion, Line 401ff: The authors used the TMN classification to stratify the patient collection. Is the expression of the exons X1, X3, X8, and X9 associated with specific subtypes of colorectal tumors, e.g. CIN, MIS, CIMP? Please add a corresponding statement to the discussion of the manuscript.
We agree with the Reviewer that analyzing the associations of DDC novel exons with other molecular characteristics of the CRC patients would be very interesting. Nevertheless, such data are not available for our cohort, unfortunately. Therefore, we decided to comment on this interesting aspect at the end of the Discussion section:
Page 19, lines 508-515 (Discussion): Since our knowledge regarding the molecular background of CRC is evolving and the significance of global genomic and epigenomic status in the initiation and progression of this malignancy is undoubtful, molecular classification of CRC patients has been adopted in clinical practice. Particularly, microsatellite instability (MSI) status, chromosomal instability (CIN) status, and CpG island methylator phenotype (CIMP) status play a significant role in determining clinical, pathological, and biological characteristics of CRC [51]. Therefore, it would be interesting to evaluate the potential of these DDC novel exons as biomarkers in subgroups of CRC patients with distinct genomic and epigenomic status.
Moreover, we included the following reference:
51.Ogino, S.; Goel, A. Molecular classification and correlates in colorectal cancer. J Mol Diagn 2008, 10, 13-27, doi:10.2353/jmoldx.2008.070082.
The authors wish to thank the Reviewers for their constructive comments that led to the improvement of the current manuscript.
